# In-hospital mortality among immunosuppressed patients with COVID-19: Analysis from a national cohort in Spain

Inés Suárez-García[1,2]☯*, Isabel Perales-Fraile[2,3]☯*, Andrés González-García[4], Arturo Muñoz-Blanco[5], Luis Manzano[6], Martín Fabregate[6], Jesús Díez-Manglano[7], Eva Fonseca Aizpuru[8], Francisco Arnalich Fernández[9], Alejandra García García[10], Ricardo Gómez-Huelgas[11], José-Manuel Ramos-Rincón[12], on behalf of SEMI-COVID-19 Network[¶]

1 Infectious Diseases Group, Department of Internal Medicine, Hospital Universitario Infanta Sofía, FIIB HUIS HHEN, Madrid, Spain, 2 Facultad de Ciencias Biomédicas y de la Salud, Universidad Europea, Madrid, Spain, 3 Department of Internal Medicine, Hospital Universitario Infanta Sofía, FIIB HUIS HHEN, Madrid, Spain, 4 Unidad de Enfermedades Sistémicas Autoinmunes y Minoritarias, Servicio de Medicina Interna, IRYCIS, Hospital Universitario Ramón y Cajal, Madrid, Spain, 5 Department of Internal Medicine, Hospital Universitario Infanta Sofía, Madrid, Spain, 6 Servicio de Medicina Interna, IRYCIS, Hospital Universitario Ramón y Cajal, Madrid, Spain, 7 Servicio de Medicina Interna, Hospital Royo Vilanova, Zaragoza, Spain, 8 Servicio de Medicina Interna, Hospital de Cabueñes, Gijón, Asturias, Spain, 9 Servicio de Medicina Interna, Hospital Universitario La Paz, Madrid, Spain, 10 Servicio de Medicina Interna, Hospital General Universitario Gregorio Marañón, Madrid, Spain, 11 Internal Medicine Department, Regional University Hospital of Málaga, Biomedical Research Institute of Málaga (IBIMA), University of Málaga (UMA), Málaga, Spain, 12 Department of Clinical Medicine, Miguel Hernandez University of Elche, Alicante, Spain

☯ These authors contributed equally to this work.
¶ Membership of the SEMI-COVID-19 Network is listed in the S1 Appendix.
* inessuarez@hotmail.com (ISG); isabel.perales@salud.madrid.org (IPF)

**Data Availability Statement:** There are ethical and legal restrictions on uploading the data. The data set contains potentially identifying information, and

## Abstract

### Background

Whether immunosuppressed (IS) patients have a worse prognosis of COVID-19 compared to non-IS patients is not known. The aim of this study was to evaluate the clinical characteristics and outcome of IS patients hospitalized with COVID-19 compared to non-IS patients.

### Methods

We designed a retrospective cohort study. We included all patients hospitalized with laboratory-confirmed COVID-19 from the SEMI-COVID-19 Registry, a large multicentre national cohort in Spain, from March 27th until June 19th, 2020. We used multivariable logistic regression to assess the adjusted odds ratios (aOR) of in-hospital death among IS compared to non-IS patients.

### Results

Among 13 206 included patients, 2 111 (16.0%) were IS. A total of 166 (1.3%) patients had solid organ (SO) transplant, 1081 (8.2%) had SO neoplasia, 332 (2.5%) had hematologic neoplasia, and 570 (4.3%), 183 (1.4%) and 394 (3.0%) were receiving systemic steroids,

the restriction was imposed by the Provincial Research Ethics Committee of Málaga, Spain. The whole data of this study are available from the study coordinators, Profs Ricardo Gomez-Huelgas and José Manuel Casa Rojo, upon reasonable request, with agreement with Spanish Law 14/2007, of July 3, on Biomedical Research; Regulation (EU) 2016/679 of the European Parliament and of the Council of 27 April 2016 on the protection of natural persons with regard to the processing of personal data and on the free movement of such data, and repealing Directive 95/46/EC (General Data Protection Regulation); and Spanish Organic Law 3/2018, of December 5, on the Protection of Personal Data and the Guarantee of Digital Rights, as well with the previous autorization by the Provincial Research Ethics Committee of Málaga (Spain). Data requests may be sent to the Spanish Society of Internal Medicine (semi@fesemi.org), Prof Ricardo Gomez-Huelgas (rgh@uma.es) and to Jose Manuel Casas-Rojo (jm.casas@gmail.com).

**Funding:** The authors received no specific funding for this work.

**Competing interests:** The authors have declared that no competing interests exist.

biological treatments, and immunosuppressors, respectively. Compared to non-IS patients, the aOR (95% CI) for in-hospital death was 1.60 (1.43–1.79) for all IS patients, 1.39 (1.18–1.63) for patients with SO cancer, 2.31 (1.76–3.03) for patients with haematological cancer and 3.12 (2.23–4.36) for patients with SO transplant. The aOR (95% CI) for death for patients who were receiving systemic steroids, biological treatments and immunosuppressors compared to non-IS patients were 2.16 (1.80–2.61), 1.97 (1.33–2.91) and 2.06 (1.64–2.60), respectively. IS patients had a higher odds than non-IS patients of in-hospital acute respiratory distress syndrome, heart failure, myocarditis, thromboembolic disease and multiorgan failure.

## Conclusions

IS patients hospitalized with COVID-19 have a higher odds of in-hospital complications and death compared to non-IS patients.

## Introduction

Immune suppression is a major condition associated with a high risk of serious infectious. Common viral agents, such as influenza, adenovirus, rhinovirus and respiratory syncytial virus, usually cause severe disease in immunosuppressed patients [1]; however, whether novel severe acute respiratory syndrome coronavirus 2 (SARS- CoV-2) has a more severe course among immunosuppressed patients is still unclear.

Demographic factors such as advanced age or male sex, as well as several conditions such as hypertension, obesity, diabetes mellitus or cardiovascular diseases, have been described as risk factors associated with adverse outcomes of coronavirus disease 2019 (COVID-19) [2, 3], but there is little evidence about its course among immunosuppressed patients. Patients with immune suppression (such as those with cancer, transplant recipients, or receiving immunosuppressive drugs) could be presumed to have a worse prognosis of COVID-19. However, in the earlier published COVID-19 series, there was a very low proportion of patients with these conditions [4–6]. Since then, studies assessing the prognosis of COVID-19 among patients with several immunosuppressive conditions have shown conflicting results. A meta-analysis conducted in China did not find a statistically significant risk of severe COVID-19 among immunosuppressed (IS) patients [7]. A systematic review including 16 articles could only gather 110 IS patients with COVID-19 and concluded that the proportion of IS patients was low compared to the overall figures of patients affected with COVID-19, and that IS children and adults seemed to have a favorable course of the disease [8]. Regarding cancer patients with COVID-19, recent reports have shown a high mortality rate (20%) [9] and a 3.5 fold increase in the risk of death, admission to the intensive care unit (ICU) and need of invasive ventilation [10]. On the other hand, a study by Miyashita *et al* [11] did not find any significant differences in COVID-19 mortality among 334 patients with cancer compared to those without cancer.

It has been hypothesized that lung damage associated with SARS-CoV-2 may be rather caused by an exaggerated immune response than by the virus itself. Cytokine storm may contribute to the pathogenesis of COVID-19 [12, 13] and lead to multiorgan failure, respiratory distress syndrome (ARDS) and death [14, 15]. The use of corticosteroids has been shown to decrease mortality due to severe COVID-19 [16], and several immunomodulatory agents are being studied in order to reduce systemic inflammation [17–19]. Whether immunosuppressed

patients may experience more severe forms of the disease due to their impaired immune response, or they may have a milder course due to a lower probability of experiencing a cyto-kine release syndrome, is still not well understood.

Given the paucity of data on the clinical presentation and the prognosis of COVID-19 among IS patients, and the conflicting results of the published studies regarding their out-comes, we designed a study that aimed to evaluate the clinical characteristics and outcome of IS patients hospitalized with COVID-19 compared to patients without immune suppression in a large retrospective national cohort in Spain.

## Methods

### Patient selection

Patients were selected from the SEMI-COVID Registry. The SEMI-COVID-19 Network is a multicentre registry developed by the Spanish Society of Internal Medicine (SEMI) that pro-vides data on the clinical characteristics, epidemiology and treatment of patients with labora-tory-confirmed COVID-19 hospitalized in Spain; details of the registry have been described elsewhere [20]. COVID-19 was confirmed in all patients either by a positive real-time poly-merase chain reaction (RT-PCR) testing of a nasopharyngeal or sputum sample, or by a posi-tive result on serological testing and compatible clinical presentation. All patients registered from March 27th until June 19th, 2020, and who had complete information on June 19th, 2020, were included.

### Variables

Patients were classified as immunosuppressed (IS) if they had had solid organ (SO) transplan-tation, active SO malignant neoplasia (with or without metastases), active haematological neo-plasia (lymphoma or leukaemia), or if they were treated with any immune suppressive treatment on a chronic basis prior to admission, including classical immunosuppressive agents (cytotoxic agents such as calcineurin inhibitors, purine analogues, folate antagonists, alkylating agents, inosine monophosphate inhibitors, mTOR inhibitors and janus-kinase inhibitors), bio-logical treatments, or systemic corticosteroids. Patients were classified as non-immunosup-pressed (non-IS) if they fulfilled none of these conditions.

In addition, we collected data on the following variables: date of admission, age, sex, smok-ing status (never smoked/ex-smoker/currently smoking), obesity, dependency level (catego-rized according to Barthel index: no/mild [>90], moderate [61–90], or severe [= <60]), Charlson comorbidity index, comorbidities (arterial hypertension, chronic heart failure, chronic obstructive bronchopulmonary disease, asthma, dementia, moderate-severe chronic liver disease [defined as chronic liver disease with portal hypertension, past or present ascites, esophageal varices, or encephalopathy], moderate-severe chronic renal failure [defined as a serum creatinine level >3 mg/dl prior to admission or history of dialysis]), diabetes mellitus, presenting symptoms at admission, laboratory tests at admission (haemoglobin, leukocyte, lymphocyte and eosinophil count, lactate dehydrogenase, D-dimer, C-reactive protein, serum creatinine), chest radiography at admission (alveolar infiltrates, interstitial infiltrates, and pleu-ral effusion), date of discharge, and in-hospital death.

The primary outcome was in-hospital mortality. We also analysed as secondary outcomes: 1. a composite index of admission to intensive care unit (ICU) or in-hospital death, 2. length of hospital stay, and 3. In-hospital complications (bacterial pneumonia, acute respiratory dis-tress syndrome [ARDS], acute heart failure, arrythmia, acute myocardial infarction, myocardi-tis, seizure, stroke, shock, acute renal failure, sepsis, disseminated intravascular coagulation

[DIC], venous thromboembolism, multiorgan failure, and acute limb ischemia). All in-hospital complications were categorized as dichotomous (yes or no).

## Statistical analysis

Descriptive analyses were carried out using frequency distributions for categorical variables and mean (standard deviation [SD]) or median (interquartile range [IQR]), as appropriate. Differences in proportions were assessed using Pearson's chi-square test and differences in means or medians (including the analysis of length of stay) were assessed using Student's t-test or Mann-Whitney's U test, as appropriate. Multivariable logistic regression models were fitted to estimate the odds of in-hospital death (and the composite outcome of admission to ICU or death) among IS compared to non-IS patients, as well as the odds of in-hospital mortality among patients with cancer (SO and haematologic), with SO transplant, and receiving immunosuppressive treatments, compared to non-immunosuppressed patients. All models were adjusted for potential confounders (age, sex, level of dependency, smoking status, arterial hypertension, chronic heart failure, chronic obstructive bronchopulmonary disease, asthma, dementia, moderate-severe chronic liver disease, moderate-severe chronic renal failure, diabetes mellitus), which were selected a priori, based on previous literature. In order to investigate whether the administration of in-hospital steroids could have influenced our mortality estimates, we repeated all multivariable analyses for the mortality primary outcome, adjusting for in-hospital steroid use in addition to all other variables. Robust methods were used to estimate confidence intervals (CI), assuming correlation between the subjects in each centre. All analyses were performed with a 95% confidence level. Statistical analyses were performed in Stata version 15 (StataCorp, College Station, TX, USA).

## Ethics

Informed consent was obtained from all the patients. When it was not possible to obtain informed consent in writing due to biosafety concerns or if the patient had already been discharged, informed consent was requested verbally and noted on the medical record.

Personal data was processed in strict compliance with Spanish Law 14/2007, of July 3, on Biomedical Research; Regulation (EU) 2016/679 of the European Parliament and of the Council of 27 April 2016 on the protection of natural persons with regard to the processing of personal data and on the free movement of such data, and Directive 95/46/EC (General Data Protection Regulation); and Spanish Organic Law 3/2018, of December 5, on the Protection of Personal Data and the Guarantee of Digital Rights. The SEMI-COVID-19 Registry was approved by the Provincial Research Ethics Committee of Malaga (Spain) on March 26, 2020.

## Results

By June 19[th], 2020, 14 599 patients were included in the SEMI-COVID Registry. Of these, 1 393 patients were excluded because they did not have valid data (they had missing data on age, date of symptom onset, date of discharge, in-hospital death, previous treatment, or confirmed positive SARS-2-CoV polymerase chain reaction). Finally, 13 206 patients were included with valid information, of which 2 111 (16.0%) were IS. Among the IS patients, 166 (1.3%) had SO transplant (118 had renal, 22 had liver, 11 had heart, and 15 had lung transplant). 1081 (8.2%) patients had SO neoplasia, of whom 276 had metastases. A total of 332 (2.5%) patients had hematologic neoplasia: 164 had leukaemia, 190 had lymphoma, and 4 patients had concomitant leukaemia and lymphoma.

Regarding immunosuppressive treatment prior to admission, 570 (4.3%) patients were receiving systemic steroids, 183 (1.4%) were receiving biological treatments, and 394 (3.0%)

were receiving immune suppressors (of which 62 patients were receiving azathioprine, 122 were receiving methotrexate, 97 were receiving tacrolimus, 11 were receiving cyclophospha-mide, 95 receiving mycophenolate, 17 were receiving cyclosporin, 61 were receiving rapamycin, 25 were receiving everolimus, and 2 were receiving Janus-kinase inhibitors [tofacitinib]).

Baseline demographic and clinical characteristics of the patients according to immune sta-tus are shown in Table 1. IS patients were on average 5 years older than the non-IS ones; they were more frequently ex- or current smokers, were more frequently moderately dependent, had higher scores in the Charlson comorbidity index, and had more frequent comorbidities (arterial hypertension, chronic heart failure, chronic obstructive bronchopulmonary disease, asthma, moderate to severe chronic liver disease, moderate to severe chronic renal failure, and diabetes mellitus).

Table 2 shows presenting symptoms, along with laboratory parameters at admission, according to immune status. Presenting symptoms were similar in both groups, although IS patients seemed to have a lower frequency of arthromyalgias at presentation. Lymphocyte count was on average 448 cells/mm$^3$ higher among IS patients; this difference was statistically significant. Chest radiography findings at presentation according to immune status are also shown in Table 2.

## Primary outcome (in-hospital death)

The proportions of patients who died during their hospital stay among IS and non-IS patients, and also among patients with SO cancer, hematologic cancer and SO transplant are shown in Table 3. Compared to non-IS patients, IS patients had a significantly higher odds of in-hospital death; this odds was still significantly higher after adjusting for other risk factors (adjusted odds ratio [aOR]: 1.60; 95% CI: 1.43–1.79).

After adjusting for other risk factors, patients with SO cancer (both metastatic and non-metastatic), haematological cancers (both lymphoma and leukaemia), or SO transplant, had a significantly higher odds of in-hospital death compared to non-IS patients. Likewise, patients receiving treatments with a suppressive effect on the immune system prior to admission (sys-temic steroids, biological treatments, or immune suppressors) had a significantly higher odds of in-hospital death compared to non-IS patients (Table 3).

IS patients were more likely to be treated with systemic steroids during their hospital stay than non-IS patients: 877 (42.0%) IS patients received in-hospital steroids compared to 3623 (32.9%) non-IS patients (p<0.001). This difference was less marked, but still significant, when considering only the 12545 patients that were not receiving steroids prior to admission: 551 (36.2%) IS versus 3623 (32.9%) non-IS patients received in-hospital steroids (p = 0.011). We repeated all multivariable analyses for the mortality primary outcome adjusting for in-hospital steroid use in addition to all other variables: our mortality estimates were not significantly changed (S1 Table).

## Secondary outcomes

IS patients were less likely to be admitted to ICU than non-IS patients (141 [6.7%] versus 953 [8.6%]; p = 0.003). Among the 1094 patients that were admitted to the ICU, the odds of death was still significantly higher among IS than among non-IS patients in univariable (OR: 2.25; 95% CI: 1.65–3.06) and multivariable (aOR: 1.91; 95% CI: 1.29–2.81) analyses.

A total of 718 (34.0%) IS and 2719 (24.5%) non-IS patients were admitted to ICU or died during the hospital stay. Compared to non-IS patients, IS patients had a significantly higher odds of admission to ICU or death (OR: 1.96; 95% CI: 1.75–2.20); this odds was still significant

**Table 1. Baseline demographic and clinical characteristics of the patients according to immune status, and of patients with solid organ cancer, haematologic cancer and solid organ transplant.**

| | Non-IS | IS | P | SO cancer | Haematologic cancer | SO transplant |
|---|---|---|---|---|---|---|
| Sex | | | | | | |
| Male | 6253 (56.4) | 1253 (59.4) | 0.014 | 695 (64.3) | 227 (63.4) | 99 (59.6) |
| Female | 4831 (43.5) | 854 (40.4) | | 383 (35.4) | 131 (36.6) | 66 (39.8) |
| Unknown | 11 (0.1) | 4 (0.2) | | 3 (0.3) | 0 (0) | |
| Age (years): mean (SD) | 66.5 (16.6) | 71.0 (13.6) | <0.001 | 73.2 (12.3) | 71.0 (14.1) | 63.5 (13.3) |
| Smoking status | | | | | | |
| Never | 7553 (68.1) | 1192 (56.5) | <0.001 | 565 (52.3) | 214 (59.8) | 105 (63.2) |
| Former | 2481 (22.4) | 676 (32.0) | | 382 (35.3) | 106 (29.6) | 50 (30.1) |
| Current | 534 (4.8) | 142 (6.7) | | 87 (8.0) | 24 (6.7) | 5 (3.0) |
| Unknown | 527 (4.7) | 101 (4.8) | | 47 (4.3) | 14 (3.9) | 6 (3.6) |
| Obesity | | | | | | |
| No | 7910 (71.3) | 1547 (73.3) | 0.127 | 813 (75.2) | 262 (73.2) | 131 (78.9) |
| Yes | 2180 (19.6) | 376 (17.8) | | 179 (16.6) | 61 (17.0) | 32 (19.3) |
| Unknown | 1005 (9.1) | 188 (8.9) | | 89 (8.2) | 35 (9.8) | 3 (1.8) |
| Dependency level | | | | | | |
| No/mild | 9203 (82.9) | 1677 (79.4) | <0.001 | 833 (77.1) | 286 (79.9) | 144 (86.7) |
| Moderate | 944 (8.5) | 282 (13.4) | | 159 (14.7) | 45 (12.6) | 15 (9.0) |
| Severe | 808 (7.3) | 123 (5.8) | | 73 (6.7) | 22 (6.1) | 2 (2.4) |
| Unknown | 140 (1.3) | 29 (1.4) | | 16 (1.5) | 5 (1.4) | 3 (1.8) |
| Charlson comorbidity index | | | | | | |
| 0–1 | 8484 (76.5) | 378 (17.9) | <0.001 | 0 (0) | 1 (0.3) | 51 (30.7) |
| 2–3 | 1713 (15.4) | 885 (41.9) | | 455 (42.1) | 216 (60.3) | 57 (34.3) |
| 4–5 | 431 (3.9) | 348 (16.5) | | 220 (20.3) | 72 (20.1) | 30 (18.1) |
| > = 6 | 182 (1.6) | 425 (20.1) | | 363 (33.6) | 58 (16.2) | 25 (15.1) |
| Unknown | 285 (2.6) | 75 (3.6) | | 43 (4.0) | 11 (3.1) | 3 (1.8) |
| Comorbidities* | | | | | | |
| Arterial hypertension | 5434 (49.0) | 1239 (58.7) | <0.001 | 648 (59.9) | 198 (55.3) | 120 (72.3) |
| Chronic heart failure | 726 (6.5) | 217 (11.3) | <0.001 | 107 (9.9) | 42 (11.7) | 17 (10.2) |
| COPD | 688 (6.2) | 236 (11.2) | <0.001 | 136 (12.6) | 31 (8.7) | 9 (5.4) |
| Asthma | 820 (7.4) | 145 (6.9) | 0.338 | 58 (5.4) | 16 (4.5) | 9 (5.4) |
| Dementia | 1120 (10.1) | 190 (9.0) | 0.131 | 109 (10.1) | 26 (7.3) | 7 (4.2) |
| Chronic liver disease** | 79 (0.7) | 55 (2.6) | <0.001 | 31 (2.9) | 5 (1.4) | 14 (8.4) |
| Chronic renal failure** | 554 (5.0) | 245 (11.6) | <0.001 | 110 (10.2) | 32 (8.9) | 78 (47.0) |
| Diabetes mellitus | 2017 (18.2) | 509 (24.1) | <0.001 | 263 (24.3) | 82 (22.9) | 57 (34.3) |
| Total | 11095 | 2111 | | 1081 | 358 | 166 |

Values are shown as n (%) unless stated otherwise. Percentages might not add up to 100% due to rounding. Non-IS: non-immunosuppressed patients. IS: immunosuppressed patients. COPD: chronic obstructive bronchopulmonary disease.

*Comorbidity values are not mutually exclusive (any given patient could have several comorbidities).

**Moderate to severe.

P-values for differences in proportions between immunosuppressed and non-immunosuppressed groups are also shown.

after adjusting for other risk factors (aOR: 1.73; 95% CI: 1.53–1.96). The length of hospital stay was significantly longer for IS than for non-IS patients: the median length of stay was 10 (IQR: 6–16) versus 9 (IQR: 6–16) days in IS and non-IS patients, respectively (p<0.001).

In-hospital complications according to immune status are shown in Table 4. After adjusting for other risk factors, IS patients had a higher odds of developing bacterial pneumonia, acute

**Table 2. Presenting symptoms, laboratory parameters and chest radiography findings at admission according to immune status.**

| | Non-IS | IS | P |
|---|---|---|---|
| **Presenting symptoms** | | | |
| Cough | 8257 (74.4) | 1482 (70.2) | <0.001 |
| Arthromyalgias | 3427 (30.9) | 505 (23.9) | <0.001 |
| Ageusia | 802 (7.2) | 121 (5.7) | 0.014 |
| Anosmia | 702 (6.3) | 121 (5.7) | 0.326 |
| Asthenia | 4726 (42.6) | 905 (42.9) | 0.829 |
| Anorexia | 2078 (18.7) | 451 (21.4) | 0.005 |
| Sore throat | 1087 (9.8) | 173 (8.2) | 0.021 |
| Headache | 1284 (11.6) | 188 (8.9) | <0.001 |
| Fever (>38˚C) | 7112 (64.1) | 1291 (61.2) | 0.010 |
| Dyspnea | 6387 (57.6) | 1177 (55.8) | 0.125 |
| Diarrhoea | 2607 (23.5) | 432 (20.5) | 0.002 |
| Nausea | 1369 (12.3) | 225 (10.7) | 0.031 |
| Vomiting | 821 (7.4) | 146 (6.9) | 0.466 |
| Abdominal pain | 703 (6.3) | 140 (6.6) | 0.594 |
| **Laboratory tests on admission** | | | |
| Haemoglobin (g/dl) | 13.9 (1.8) | 13.0 (2.1) | <0.001 |
| Leukocyte count (cells/mm3) | 7162 (4620) | 8165 (8900) | <0.001 |
| Lymphocyte count (cells/mm3) | 1096 (1038) | 1543 (5011) | <0.001 |
| Eosinophil count (cells/mm3) | 35.8 (124.4) | 42.4 (176.2) | 0.038 |
| Lactate dehydrogenase (U/l) | 365 (205) | 373 (260) | 0.142 |
| Ferritin (microg/l) | 942 (1081) | 942 (1129) | 0.998 |
| D-dimer (ng/ml) | 1670 (7727) | 2119 (5604) | 0.030 |
| C-reactive protein (mg/l) | 85 (87) | 91 (90) | 0.012 |
| Creatinine (mg/dl) | 1.07 (0.81) | 1.26 (1.08) | <0.001 |
| **Chest radiography on admission** | | | |
| Alveolar infiltrates | 5345 (48.2) | 1031 (48.8) | 0.585 |
| Interstitial infiltrates | 6945 (62.6) | 1175 (55.6) | <0.001 |
| Pleural effusion | 444 (4.0) | 147 (7.0) | <0.001 |

Values for presenting symptoms and chest radiography on admission are shown as n (%), and values for laboratory tests on admission are shown as mean (standard deviation). Non-IS: non-immunosuppressed patients. IS: immunosuppressed patients.

respiratory distress syndrome, heart failure, myocarditis, thromboembolic disease, and multi-organ failure compared to non-IS patients.

## Discussion

Our study shows that IS patients with COVID-19 had a higher odds of in-hospital death compared to non-IS patients. The odds of death was also higher than that of non-IS patients when considering each of the IS patient subgroups: active SO cancer (whether metastatic or not), hematologic cancer, SO transplant or use of immunosuppressive drugs. This study has analyzed, to our knowledge, the largest number of IS patients with COVID-19 published to date. Data on the clinical presentation and prognosis of COVID-19 among immune suppressed patients is very scarce in the literature. A meta-analysis by Gao et al [7] could only include 5

**Table 3. Number of deaths (%), total number of patients, and crude and adjusted OR for death among immunosuppressed patients, patients with specific diseases or conditions (cancer [solid organ or haematologic], solid organ transplant or systemic autoimmune diseases), and patients receiving immune suppressive treatments prior to admission (systemic steroids, biological treatments, or immunosuppressors).** All analyses have use non-immunosuppressed patients as reference category.

| | Deaths: n (%) | N | OR (95% CI) | aOR* (95% CI) | p* |
|---|---|---|---|---|---|
| Non-IS | 2143 (19.3) | 11095 | 1 | 1 | |
| IS | 661 (31.3) | 2111 | 1.90 (1.72–2.11) | 1.60 (1.43–1.79) | <0.001 |
| Patients with specific diseases and conditions | | | | | |
| All cancers (SO and H) | 465 (33.3) | 1398 | 2.08 (1.81–2.38) | 1.59 (1.38–1.82) | <0.001 |
| SO cancer | 343 (31.7) | 1081 | 1.94 (1.66–2.27) | 1.39 (1.18–1.63) | <0.001 |
| SO cancer with MT | 84 (30.4) | 276 | 1.82 (1.37–2.43) | 1.87 (1.33–2.63) | <0.001 |
| SO cancer, no MT | 259 (32.2) | 805 | 1.98 (1.63–2.41) | 1.27 (1.05–1.54) | 0.013 |
| Hematologic cancer | 139 (38.8) | 358 | 2.42 (1.92–3.05) | 2.31 (1.76–3.03) | <0.001 |
| Leukaemia | 66 (39.3) | 168 | 2.70 (1.89–3.84) | 2.20 (1.49–3.25) | <0.001 |
| Lymphoma | 77 (40.0) | 194 | 2.75 (2.16–3.51) | 2.94 (2.19–3.95) | <0.001 |
| Transplant | 57 (34.3) | 166 | 2.18 (1.60–2.99) | 3.12 (2.23–4.36) | <0.001 |
| Patients receiving immune suppressive treatments prior to admission | | | | | |
| Systemic steroids | 202 (35.4) | 570 | 2.29 (1.96–2.68) | 2.16 (1.80–2.61) | <0.001 |
| Biological treatment | 49 (26.8) | 183 | 1.52 (1.06–2.19) | 1.97 (1.33–2.91) | 0.001 |
| Immunosuppressors** | 109 (27.7) | 394 | 1.59 (1.27–1.99) | 2.06 (1.64–2.60) | <0.001 |

Non-IS: non-immunosuppressed patients. IS: immunosuppressed patients. SAID: systemic autoimmune diseases. OR: crude odds ratio. CI: confidence interval. aOR: adjusted odds ratio. SO: solid organ. H: haematological. MT: metastases.

*Adjusted for age, sex, level of dependency, smoking status, and comorbidities (arterial hypertension, chronic heart failure, chronic obstructive bronchopulmonary disease, asthma, dementia, moderate-severe chronic liver disease, moderate-severe chronic renal failure, and diabetes mellitus).

**Immunosuppressors include: azathioprine, methotrexate, tacrolimus, cyclophosphamide, mycophenolate, cyclosporin, rapamycin, and everolimus.

**Table 4. In-hospital complications according to immune status: Number of patients developing in-hospital complications, and crude and adjusted OR compared to non-immunosuppressed patients.**

| | Non-IS | IS | OR (95% CI) | aOR* (95% CI) | p* |
|---|---|---|---|---|---|
| Bacterial pneumonia | 1155 (10.4) | 278 (13.2) | 1.31 (1.12–1.52) | 1.17 (1.00–1.35) | 0.038 |
| ARDS | 3527 (31.8) | 820 (38.8) | 1.33 (1.21–1.53) | 1.18 (1.05–1.33) | 0.006 |
| Acute heart failure | 586 (5.3) | 185 (8.8) | 1.72 (1.41–2.10) | 1.35 (1.09–1.68) | 0.006 |
| Arrythmia | 408 (3.7) | 114 (5.4) | 1.50 (1.18–1.90) | 1.22 (0.96–1.56) | 0.111 |
| Acute myocardial infarction | 92 (0.8) | 16 (0.8) | 0.91 (0.55–1.52) | 0.73 (0.45–1.20) | 0.215 |
| Myocarditis | 91 (0.8) | 31 (1.5) | 1.80 (1.20–2.71) | 1.61 (1.07–2.41) | 0.022 |
| Seizure | 61 (0.5) | 17 (0.8) | 1.46 (0.82–2.64) | 1.32 (0.71–2.46) | 0.383 |
| Stroke | 72 (0.6) | 16 (0.8) | 1.17 (0.63–2.18) | 0.98 (0.53–1.83) | 0.958 |
| Shock | 498 (4.5) | 107 (5.1) | 1.14 (0.92–1.40) | 1.02 (0.83–1.27) | 0.825 |
| Acute renal failure | 1475 (13.3) | 390 (18.5) | 1.48 (1.25–1.75) | 1.14 (0.98–1.34) | 0.094 |
| Sepsis | 679 (6.1) | 160 (7.6) | 1.26 (1.06–1.49) | 1.11 (0.93–1.32) | 0.256 |
| DIC | 111 (1.0) | 38 (1.8) | 1.81 (1.19–2.75) | 1.55 (0.99–2.42) | 0.056 |
| Venous thromboembolism | 208 (1.9) | 62 (2.9) | 1.59 (1.29–1.95) | 1.61 (1.29–2.01) | <0.001 |
| Multiorgan failure | 638 (5.8) | 203 (9.6) | 1.74(1.52–2.00) | 1.41 (1.22–1.64) | <0.001 |
| Acute limb ischemia | 53 (0.5) | 12 (0.6) | 1.19 (0.66–2.16) | 1.13 (0.61–2.11) | 0.692 |

Values are shown as n (%). Non-IS: non-immunosuppressed patients. IS: immunosuppressed patients. OR: odds ratio. CI: confidence interval. ARDS: acute respiratory distress syndrome. DIC: disseminated intravascular coagulation.

*Adjusted for age, sex, level of dependency, smoking status, and comorbidities (arterial hypertension, chronic heart failure, chronic obstructive bronchopulmonary disease, asthma, dementia, moderate-severe chronic liver disease, moderate-severe chronic renal failure, and diabetes mellitus).

studies involving 776 patients with immunosuppression and suggested a higher risk of severe COVID-19 compared to non-IS patients, but this difference was not statistically significant.

Immunosuppression was associated with a more severe course of COVID-19 in our study: we identified a higher odds of in-hospital death, in-hospital death or ICU admission, and several in-hospital complications (bacterial pneumonia, ADRS, heart failure, myocarditis, thromboembolic disease, and multiorgan failure). We also found a longer length of stay among IS patients compared to the non-IS group. IS patients were older, had more severe dependency and had more frequent comorbidities than the non-IS ones. The mortality odds was still significantly higher among IS patients and among all the subgroups after adjusting for all these variables. However, we cannot completely exclude some residual confounding due to non-measured variables.

Despite a higher mortality odds, the clinical presentation was not very different among IS compared to non-IS patients: IS patients were less likely to present with cough, myalgias or headache than the non-IS group, but the absolute differences in these proportions were low. Similarly, IS patients were more likely than the non-IS ones to have pleural effusion and less likely to have bilateral interstitial infiltrates in the chest X-rays on admission, but the absolute differences were also low.

Regarding laboratory tests on admission, compared to non-IS patients, the IS group had higher average levels of inflammatory markers (D-dimer and C-reactive protein); these markers have been associated with a worse prognosis of COVID-19 [21]. However, lymphocyte and eosinophil counts were significantly higher among IS patients. Lymphopenia causes a defect in antiviral regulatory immunity, and lymphopenia and eosinopenia have been described as markers of severe manifestations of COVID-19 [22]. It is therefore surprising that, despite a worse prognosis, IS patients had higher lymphocyte and eosinophil counts. Other differences such as lower hemoglobin concentration and higher plasma creatinine levels are not unexpected and could be explained by the use of myelotoxic or nephrotoxic immunosuppressive or chemotherapeutic agents, digestive tract cancers, myelosuppression due to hematologic malignancies, and the proportion of renal transplant patients who could have a lower glomerular filtration rate [23].

Overall, the proportion of in-hospital deaths in the SEMI-COVID-19 cohort was very high compared to other series [3, 24], as has been described elsewhere [20]. This has been attributed not only to the older age of Spanish in-hospital patients compared to those in the earlier series [3, 25], but also to the overloaded healthcare system during the first wave of the pandemic, which might in turn have risen thresholds for hospitalization and given less chances of invasive ventilation or admission to the ICU to older patients or those with comorbidities. After adjusting for other risk factors (such as age, dependency level and comorbidities), IS patients had on average a 60% higher odds of in-hospital death than the non-IS ones. We could hypothesize that this higher odds would be due to a lower probability of being admitted to the ICU, as patients with severe comorbidities or advanced cancer are not usually candidates for intensive care. However, we believe that this does not fully explain the higher mortality in our IS patients for the following reasons: first, although the proportion of patients admitted to the ICU was significantly lower among IS patients compared to the non-IS ones, the overall proportions of patients admitted to the ICU were low in both groups, as was the absolute difference in proportions; second, the composite index of in-hospital death or admission to ICU was still significantly higher in the IS compared to the non-IS group; and third, among patients that were admitted to the ICU, mortality was still significantly higher for the IS patients.

Dexamethasone is so far the only treatment that has shown a reduction in mortality in patients with COVID-19 [16]. Therefore, we sought out to investigate whether our results could be confounded by the administration of in-hospital steroids. IS patients were more likely

to receive steroids than non-IS suppressed ones; this can be partly due to the fact that patients already receiving steroids before admission were more likely to remain with this treatment. However, IS patients that were not receiving steroids before admission were also more likely to receive in-hospital steroids. Nevertheless, our mortality estimates were not changed when adjusting for the use of in-hospital steroids.

The proportions of patients with SO and hematological cancer among all patients admitted with COVID-19 in our study were much higher than the overall cancer prevalence (including solid and hematological cancers) in the general Spanish population (1.61% in 2018) [26]. Studies from New York [11] and China [10] have also shown an increased proportion of patients with cancer among those hospitalized with COVID-19, with cancer prevalence of 6% and 1%, respectively.

Overall, a third of the patients with cancer (solid or hematological) who were admitted with COVID-19 died in the hospital. Compared to non-IS patients, the odds of death was higher for patients with SO cancer (whether metastatic or not), and with hematological cancer (leukemia and lymphoma). As expected, the odds of death among patients with SO cancer was higher in those with metastatic disease. Contrary to our findings, a previous study by Dai *et al* [27] did not find any significant differences in the risk of death from COVID-19 of patients with non-metastatic cancer and those without cancer; however, this was probably due to the lower sample size of his study. Previous studies have shown mortality rates between 9% and 33% among patients with cancer and COVID-19 [28, 29], with higher case-fatality rates for those with hematological malignancies.

Not all cancer patients should be considered equally immunocompromised, but especially those on active chemotherapy and those with hematological tumors such as leukemia, lymphoma, multiple myeloma and myelodysplastic syndromes [1]. Worse performance status [30], active cancer treatment, and having received chemotherapy within four weeks before admission [9], have also been related with a poorer prognosis, as well as being affected by lung cancer or a hematological neoplasm [9, 28, 31, 32]. Unfortunately, as the SEMI-COVID-19 Registry was not specifically designed to evaluate cancer patients, we did not have information such as the primary SO cancer location, time since cancer diagnosis, or whether the patients were receiving chemotherapy. Also, the only hematological neoplasms recorded were leukemias and lymphomas, and other hematologic diseases such as multiple myeloma or myelodysplastic syndromes were not registered.

To our knowledge, our study includes the highest number of SO transplant patients with COVID-19 published to date; most of these had received renal transplant. Patients with SO transplant were younger and had lower dependency level than the non-IS patients; despite this, their mortality odds was much higher. Most reports on COVID-19 transplant patients have been limited to small case series with conflicting results: while some have shown similar outcomes to the general population [33, 34], others reported a very high mortality, as found by our study [35, 36]. The largest study of SO transplant recipients reported data on 90 patients, of which 68 (76%) were hospitalized, with an in-hospital mortality of 24% [35]. Another study analyzed 24 patients with kidney transplant, 41.6% of which died [36]. We could not assess whether the prognosis would differ according to the time after transplantation as this variable was not registered in the database.

Although several immune suppressors and biological drugs have been proposed to modulate the cytokine release during severe COVID-19 [13, 37], and corticosteroids have shown to decrease mortality among patients with severe COVID-19 [16], our study found that these agents were associated with a higher mortality, at least when taken in a chronic basis prior to admission. With the exception of corticosteroids, we could not determine whether immunosuppressive drugs were maintained during hospital admission, as only prior treatments were

registered. Also, we could not analyse the effect of corticosteroid dose on our outcomes, as the doses were not registered in the SEMI-COVID-19 database. However, even low doses of corticosteroids can impair the function of the immune system when taken on a chronic basis [38].

Our study has several limitations. It is an observational retrospective study that was designed to describe the evolution of patients hospitalized with COVID-19, and it was not specifically designed to evaluate immunosuppressed patients. Therefore, some important variables such as the clinical details of each immunosuppressive condition, or whether chronic immunosuppressive treatment was maintained during hospitalization are lacking. We could not include patients with several immunosuppressive conditions such as primary immunodeficiencies, asplenia, complement deficiency or advanced HIV infection, as information about these conditions was not available in the database. Also, for patients with SO transplant there was no information on the date of the transplant, and for cancer patients there was no information on the primary tumor site, time since cancer diagnosis or chemotherapy administration. A major strength of our study is its large sample size from a multicenter national cohort, which allows us to present the largest series of IS patients, patients with cancer and patients with SO transplant published to date.

In conclusion, our study has shown that immunosuppressed patients hospitalized with COVID-19 have a higher odds of in-hospital death and several in-hospital complications than non-IS patients. The odds of in-hospital death was also higher among patients with cancer (SO or hematologic), those with SO transplant, and those who were receiving immunosuppressive medication. These groups are a vulnerable population for complicated COVID-19 and should be closely monitored.

## Supporting information

**S1 Table.**
(DOCX)

**S1 Appendix.**
(DOCX)

## Acknowledgments

We gratefully acknowledge all the investigators who participate in the SEMI-COVID-19 Registry.

## Author Contributions

**Conceptualization:** Inés Suárez-García, Isabel Perales-Fraile.

**Data curation:** Isabel Perales-Fraile, Andrés González-García, Arturo Muñoz-Blanco, Jesús Díez-Manglano, Eva Fonseca Aizpuru, Francisco Arnalich Fernández, Alejandra García García.

**Formal analysis:** Inés Suárez-García.

**Investigation:** Inés Suárez-García, Isabel Perales-Fraile.

**Methodology:** Inés Suárez-García.

**Project administration:** Ricardo Gómez-Huelgas, José-Manuel Ramos-Rincón.

**Supervision:** Inés Suárez-García.

**Writing – original draft:** Inés Suárez-García, Isabel Perales-Fraile.

**Writing – review & editing:** Inés Suárez-García, Isabel Perales-Fraile, Andrés González-García, Luis Manzano, Martín Fabregate, Jesús Díez-Manglano, Eva Fonseca Aizpuru, Francisco Arnalich Fernández, Alejandra García García, Ricardo Gómez-Huelgas, José-Manuel Ramos-Rincón.

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
