## [Decision Letter · Decision Letter 0]

23 Jun 2021

PONE-D-21-09564

In-hospital mortality among immunosuppressed patients with COVID-19: analysis from a national cohort in Spain

PLOS ONE

Dear Dr. Suárez-García,

Thank you for submitting your manuscript to PLOS ONE. After careful consideration, we feel that it has merit but does not fully meet PLOS ONE’s publication criteria as it currently stands. Therefore, we invite you to submit a revised version of the manuscript that addresses the points raised during the review process.

We look forward to receiving your revised manuscript.

Kind regards,

Aleksandar R. Zivkovic

Academic Editor

PLOS ONE

Journal Requirements:

5. One of the noted authors is a group or consortium; SEMI-COVID-19 Network.

In addition to naming the author group, please list the individual authors and affiliations within this group in the acknowledgments section of your manuscript.

Please also indicate clearly a lead author for this group along with a contact email address.

Reviewers' comments:

Reviewer #1: This manuscript is well performed, technically sound and statistically rigorous. A repeated analysis greatly appreciated to investigate influence of in hospital steroids in mortality estimates. Limitations are well stated for a retrospective study design.

The manuscript is well written and presented in proper flow.

Reviewer #2: data which is pr the presented by author is fully complying and showed the death rate is more in IS patient when they are affected by the COVID 19. this article showed the mortality rate of individuals having IS status in comparison with non IS patient. article is written well and can be taken for acceptance

Reviewer #3: 1. Summary of Research

Suárez-García et al provided a detailed breakdown of characteristics of immunosuppressed (IS) patients hospitalized with COVID-19 and assessed the association between IS status and in-hospital mortality in this cross-sectional study of data from the SEMI-COVID-19 Network. This is the largest study of IS patients with COVID-19 to date, and it showed higher odds of mortality among those with IS compared to those without IS with differences by IS condition. The results add to the literature a more detailed picture of the characteristics and prognosis of those with IS who are hospitalized with COVID-19. The manuscript would benefit from some reorganization and clarification of the methods/results, clearer definitions of exposures and some outcomes, updates to the tables, including providing more thorough footnotes, and additional minor edits.

2. Major Improvements

- Exposure classification (Immunosuppressed). The authors state in the limitations that they don’t have data on HIV/AIDS, which is why those conditions were not included in the definition of Immunosuppressed (IS). Do the authors have data on the following conditions that could also be considered as a classification for IS: Stem cell transplantation, immunoglobulin deficiency, complement deficiency, asplenia, or cytopenias or other immunodeficiencies? If not, perhaps those could be included in the limitations.

- Exposure classification (Immunosuppressed). The authors classified those with systemic corticosteroid use as IS. Systemic corticosteroid use is common treatment for COVID-19 among hospitalized patients. Do the authors have data on route, duration, or dosing of steroid therapy that might have enable classification of whether steroids were immunosuppressive? Otherwise I might suggest classifying patients with only systemic steroids as non-IS, at least for the primary analysis. A secondary/sensitivity analysis could consider those patients as IS.

- Methods, Statistical Analysis: It is not clear how the authors modeled length of hospital stay (secondary outcome #2). Did they dichotomize the length of stay (e.g. 0-2 days versus 2+ days)? Did they consider using Cox proportional hazards to model time to discharge? Either way, the methods aren't clear on how this secondary outcome was assessed. In addition, the author did not present the results of this analysis, but they briefly include it in the discussion. The authors should clarify the methods and include the results of this analysis.

- Results: I recommend including the overall mortality and mortality by each stratum to the paragraph discussing the results from Table 1. I would also add row for mortality and other outcomes to table 1, both overall and by each stratum. Authors could add a column with overall study patients, and add rows for in-hospital mortality, ICU admission, and hospital length of stay.

- Results: I also recommend including subheadings within the results section for each of the primary and secondary analyses.

3. Minor Improvements

- Methods, Study Design. The authors state the study design is retrospective cohort; however, the study design is actually cross-sectional. The exposure and outcome data among the SEMI-COVID-19 cohort are ascertained at the same time for this study, although it uses data from a cohort.

- Methods, Patient Selection. The authors could consider adding a sentence or two describing the case definition criteria for the SEMI-COVID-19 Network and whether hospitalizations included in this study are laboratory-confirmed.

- Title. If the cases are laboratory-confirmed, perhaps change the title to “In-hospital mortality among immunosuppressed patients with laboratory-confirmed COVID-19: analysis from a national cohort in Spain”.

- Methods, Statistical Analysis: Perhaps the authors could state in the methods that the in-hospital complications are dichotomized (yes/no). E.g. on line 184, they could state "3. Dichotomized (yes/no) in-hospital complications (pneumonia, ARDS, etc...."

- Methods, Statistical Analysis: The authors should include how they defined “nosocomial pneumonia” on line 1984. Also, it is unclear why authors are only interested in nosocomial pneumonia rather than all types of pneumonia.

- Results: the authors cite a cutoff date for inclusion in the study of June 19th, but in the first line of the results it says "June 16th".

- Results (Table 2): I recommend editing the footnote to be a bit more specific or adding additional rows for units in the table explaining that under laboratory tests, the first value is a mean or median (?) and the value in parentheses is SD. It is unclear as is.

- Results: In the first paragraph following Table 3 on page 17, I recommend including the aOR's and 95% confidence intervals for the models within the text of the results section.

- Results: In the first paragraph following Table 4 starting on page 18, the sentences beginning with “In order to investigate” through “in addition to all other variables” is describing methods, and as such, should be moved to the methods section. Right now, the methods and results for this analysis are being combined in the results section.

4. Additional Suggestions

- Throughout the manuscript, the authors use the term “risk,” but since the model provides an estimate of the odds ratio, then the term “odds” is more accurate. The authors should consider changing the word “risk” to “odds” throughout the manuscript.

- I would present percentages with the n’s throughout the entire manuscript. The n’s are difficult to interpret.

o All throughout the abstract, also include percentages following the n’s. (e.g. on line 90 “Among IS patients, a total of 166 (X.X%) had solid organ (SO) transplant...”)

o In the manuscript, also include the percentages. (e.g. on line 228 “Among the IS patients, 166 (XX.X%) had SO transplant…”)

o On line 233, authors report only percentages and not n’s. I would do both throughout the entire manuscript for consistency and clarity.

6. PLOS authors have the option to publish the peer review history of their article (what does this mean?). If published, this will include your full peer review and any attached files.

Reviewer #1: No

Reviewer #2: No

Reviewer #3: **Yes: **Rachel Holstein, MPH

---

## [Author Response · Author response to Decision Letter 0]

8 Jul 2021

Reviewer #1: This manuscript is well performed, technically sound and statistically rigorous. A repeated analysis greatly appreciated to investigate influence of in hospital steroids in mortality estimates. Limitations are well stated for a retrospective study design.

The manuscript is well written and presented in proper flow.

Reviewer #2: data which is pr the presented by author is fully complying and showed the death rate is more in IS patient when they are affected by the COVID 19. this article showed the mortality rate of individuals having IS status in comparison with non IS patient. article is written well and can be taken for acceptance

Reviewer #3: 1. Summary of Research

Suárez-García et al provided a detailed breakdown of characteristics of immunosuppressed (IS) patients hospitalized with COVID-19 and assessed the association between IS status and in-hospital mortality in this cross-sectional study of data from the SEMI-COVID-19 Network. This is the largest study of IS patients with COVID-19 to date, and it showed higher odds of mortality among those with IS compared to those without IS with differences by IS condition. The results add to the literature a more detailed picture of the characteristics and prognosis of those with IS who are hospitalized with COVID-19. The manuscript would benefit from some reorganization and clarification of the methods/results, clearer definitions of exposures and some outcomes, updates to the tables, including providing more thorough footnotes, and additional minor edits.

2. Major Improvements

- Exposure classification (Immunosuppressed). The authors state in the limitations that they don’t have data on HIV/AIDS, which is why those conditions were not included in the definition of Immunosuppressed (IS). Do the authors have data on the following conditions that could also be considered as a classification for IS: Stem cell transplantation, immunoglobulin deficiency, complement deficiency, asplenia, or cytopenias or other immunodeficiencies? If not, perhaps those could be included in the limitations.

Answer: Unfortunately, there is no information on these conditions in the SEMI-COVID-19 Registry. We have included this limitation in the discussion (page 24, lines 476-478).

- Exposure classification (Immunosuppressed). The authors classified those with systemic corticosteroid use as IS. Systemic corticosteroid use is common treatment for COVID-19 among hospitalized patients. Do the authors have data on route, duration, or dosing of steroid therapy that might have enable classification of whether steroids were immunosuppressive? Otherwise I might suggest classifying patients with only systemic steroids as non-IS, at least for the primary analysis. A secondary/sensitivity analysis could consider those patients as IS.

Answer: the patients that were classified as immunosuppressed were the ones who received corticosteroids prior to admission as a chronic treatment on a regular basis (we have specified this in the methods, page 8 lines 162-167); patients who only received corticosteroids during their hospital stay, or patients who were receiving inhaled or topical corticosteroids, were not included in this group. We do not have information on the dose of corticosteroids but we do know that to be included in this group the patients had to receive them via systemic route (either oral, intramuscular or intravenous) and the corticosteroids had to be received as a chronic treatment (we do not have information on the specific duration of the treatment: patients were judged to be on chronic steroid treatment by their treating physician). Chronic administration of systemic steroids even in low doses (equivalent to 5 mg/day of prednisone or lower) can impair the function of the immune system (Rheum Dis Clin North Am 2016; 42:157-176). As all these patients were receiving the steroids via systemic route and on a chronic basis, we believe that they should be included in the immunosuppressed group. We have added a comment on this in the discussion (page 24, lines 467-470)

- Methods, Statistical Analysis: It is not clear how the authors modeled length of hospital stay (secondary outcome #2). Did they dichotomize the length of stay (e.g. 0-2 days versus 2+ days)? Did they consider using Cox proportional hazards to model time to discharge? Either way, the methods aren't clear on how this secondary outcome was assessed. In addition, the author did not present the results of this analysis, but they briefly include it in the discussion. The authors should clarify the methods and include the results of this analysis.

Answer: The result for the difference in length of stay between immunosuppressed and non-immunosuppressed patients was already shown in the results (page 18, lines 330-333). Length of stay was not modelled: the values for both groups were described as median (IQR) and the p-value for the difference in medians was obtained with Mann-Whitney’s U-test. We have specified this in the methods to clarify (page 9, lines 195-196).

- Results: I recommend including the overall mortality and mortality by each stratum to the paragraph discussing the results from Table 1. I would also add row for mortality and other outcomes to table 1, both overall and by each stratum. Authors could add a column with overall study patients, and add rows for in-hospital mortality, ICU admission, and hospital length of stay.

Answer: We have added the overall mortality aOR and 95% confidence interval to the text as suggested (page 16, lines 280-281). However, for the sake of clarity and simplicity, we have not added all the aOR of mortality for each stratum as they are easy to find in table 3 and adding these to the text could make the text more difficult to read (there are 11 aOR with their corresponding 95% CI). 

Table 1 shows the baseline clinical and demographic variables for all immunosuppressed patients and for each stratum and therefore we do not think it should show the results for the mortality outcome, which are already shown in table 3 (both overall and by each stratum). Also, we do not think it is necessary to add a column with overall study patients, as the figures for all the study patients can be easily calculated by adding up the n in the columns for immunosuppressed and nonimmunosuppressed patients. The secondary outcomes ICU admission or death, and hospital length of stay, were only calculated for the overall immunosuppressed patients and their results are shown in the text (lines xx and xx, respectively)

- Results: I also recommend including subheadings within the results section for each of the primary and secondary analyses.

Answer: we have included the subheadings.

3. Minor Improvements

- Methods, Study Design. The authors state the study design is retrospective cohort; however, the study design is actually cross-sectional. The exposure and outcome data among the SEMI-COVID-19 cohort are ascertained at the same time for this study, although it uses data from a cohort.

Answer: We respectfully disagree with this. The SEMI-COVID-19 is a retrospective cohort study (Rev Clin Esp 2020; 220: 480-494). The exposure and outcome data were ascertained at different times as the exposure (immunosuppression) was ascertained first and the outcome (death) was ascertained afterwards (both were registered in the patients’ clinical records at different times). A cohort study is “defined on the basis of presence or absence of exposure to a suspected risk factor for a disease [in this case, immunosuppression] (…) eligible participants are then followed over a period of time to assess the occurrence of that outcome [in this case, death]” (Hennekens CH, Epidemiology in Medicine. 1st ed. Lippicott Williams and Wilkins. Philadelphia, 1987, p. 153). “In retrospective cohort studies (…) all the relevant events have already occurred when the study is initiated” (op. cit. p 154). Therefore, we think our study design is a retrospective cohort.

- Methods, Patient Selection. The authors could consider adding a sentence or two describing the case definition criteria for the SEMI-COVID-19 Network and whether hospitalizations included in this study are laboratory-confirmed.

Answer: All cases were laboratory-confirmed. We have added this to the methods as requested (page 8, lines 152-156), and the abstract as well.

- Title. If the cases are laboratory-confirmed, perhaps change the title to “In-hospital mortality among immunosuppressed patients with laboratory-confirmed COVID-19: analysis from a national cohort in Spain”.

Answer: We have not changed the title but we have specified that cases were laboratory-confirmed in the abstract and the methods (page 5, line 85 and page 8 lines 152-156).

- Methods, Statistical Analysis: Perhaps the authors could state in the methods that the in-hospital complications are dichotomized (yes/no). E.g. on line 184, they could state "3. Dichotomized (yes/no) in-hospital complications (pneumonia, ARDS, etc...."

Answer: we have explained it in the results section as requested (page 9, lines 188-189).

- Methods, Statistical Analysis: The authors should include how they defined “nosocomial pneumonia” on line 1984. Also, it is unclear why authors are only interested in nosocomial pneumonia rather than all types of pneumonia.

Answer: this was a mistake. What was actually registered in the database was “bacterial pneumonia”, not nosocomial pneumonia. We have changed this in the manuscript. Bacterial pneumonia was diagnosed based on a compatible clinical picture and radiographic infiltrates.

- Results: the authors cite a cutoff date for inclusion in the study of June 19th, but in the first line of the results it says "June 16th".

Answer: this was a typo and it has been corrected. It should say “June 19th”

- Results (Table 2): I recommend editing the footnote to be a bit more specific or adding additional rows for units in the table explaining that under laboratory tests, the first value is a mean or median (?) and the value in parentheses is SD. It is unclear as is.

Answer: we have specified this in the footnote as requested.

- Results: In the first paragraph following Table 3 on page 17, I recommend including the aOR's and 95% confidence intervals for the models within the text of the results section.

Answer: The aOR and 95% confidence intervals for each stratum are easy to find in table 3 and adding these to the text could make the text more difficult to read (there are 11 aOR with their corresponding 95% CI). For the sake of clarity and simplicity, we have decided not to include these values in the text and keep them in the table.

- Results: In the first paragraph following Table 4 starting on page 18, the sentences beginning with “In order to investigate” through “in addition to all other variables” is describing methods, and as such, should be moved to the methods section. Right now, the methods and results for this analysis are being combined in the results section.

Answer: we have moved this to the methods section.

4. Additional Suggestions

- Throughout the manuscript, the authors use the term “risk,” but since the model provides an estimate of the odds ratio, then the term “odds” is more accurate. The authors should consider changing the word “risk” to “odds” throughout the manuscript.

Answer: we have changed the term “risk” to “odds” throughout the manuscript as requested.

- I would present percentages with the n’s throughout the entire manuscript. The n’s are difficult to interpret.

o All throughout the abstract, also include percentages following the n’s. (e.g. on line 90 “Among IS patients, a total of 166 (X.X%) had solid organ (SO) transplant...”)

o In the manuscript, also include the percentages. (e.g. on line 228 “Among the IS patients, 166 (XX.X%) had SO transplant…”)

o On line 233, authors report only percentages and not n’s. I would do both throughout the entire manuscript for consistency and clarity.

Answer: we have added the n (%) throughout the entire manuscript as requested.

Journal Requirements:

Answer: We have checked this

Answer: We have checked this

Answer: We have added a table with this analysis as a supplementary material and eliminated the phase “data not shown” in the manuscript.

Answer: There are ethical and legal restrictions on sharing the data. The full details have been explained in the manuscript (page 25, lines 505-518). This data availability statement is the same for all the studies from the SEMI-COVID-19 Registry. A previous article from this registry, with the same data availability statement, has been previously published in PLos ONE (PLos ONE 16(2): e0247422).

5. One of the noted authors is a group or consortium; SEMI-COVID-19 Network.

In addition to naming the author group, please list the individual authors and affiliations within this group in the acknowledgments section of your manuscript.

Please also indicate clearly a lead author for this group along with a contact email address.

Answer: We have done this

---

## [Editor Report · Decision Letter 1]

19 Jul 2021

In-hospital mortality among immunosuppressed patients with COVID-19: analysis from a national cohort in Spain

PONE-D-21-09564R1

Dear Dr. Suárez-García,

We’re pleased to inform you that your manuscript has been judged scientifically suitable for publication and will be formally accepted for publication once it meets all outstanding technical requirements.

Kind regards,

Aleksandar R. Zivkovic

Academic Editor

PLOS ONE

---

## [Editor Report · Acceptance letter]

23 Jul 2021

PONE-D-21-09564R1 

In-hospital mortality among immunosuppressed patients with COVID-19: analysis from a national cohort in Spain 

Dear Dr. Suárez-García:

I'm pleased to inform you that your manuscript has been deemed suitable for publication in PLOS ONE. Congratulations! Your manuscript is now with our production department. 

Kind regards, 

on behalf of

Dr. Aleksandar R. Zivkovic 

Academic Editor

PLOS ONE